# Effect of Adding Waste Polyethylene and GGBFS on the Engineering Properties of Cement Mortar

**Chang-Chi Hung [1], Jung-Nan Chang [2], Her-Yung Wang [3],\* and Fu-Lin Wen [3]**

1   School of Architecture and Civil Engineering, Huizhou University, Huizhou 516007, China
2   Department of Tourism and Recreation Management, Fooyin University, Kaohsiung City 831, Taiwan
3   Department of Civil Engineering, National Kaohsiung University of Science and Technology, Kaohsiung City 807, Taiwan
\*   Correspondence: wangho@nkust.edu.tw; Tel.: +886-7-381-4526 (ext. 15237); Fax: +886-7-3961321

**Abstract:** The recycling of waste materials has become an important topic worldwide. Wastes can be effectively used in concrete to improve its characteristics. This study aimed to research cement mortar's physical properties, mechanical properties, and durability. In a cement mortar with a fixed water-to-binder ratio (W/B) of 0.5, waste polyethylene (PE) was added at sand volume ratios of 0%, 1%, 2%, 3%, and 4%. Cement was replaced with 0%, 10%, and 20% ground granulated blast furnace slag (GGBFS). The results showed that the slump and flow of mortar tended to decline as the added amount of waste PE increased, but they also increased with the increased replaced amount of GGBFS. The setting time of mortar was shortened as the waste PE increased but delayed as the amount of GGBFS increased. In terms of mechanical properties, the compressive strength of mortar declined as the replaced amount of waste PE increased. Using the GGBFS to replace part of the cement can improve the later mortar strength. This study found that when the added waste PE was within 2% and the replacement amount of GGBFS was 10%, the goal of recycling waste was reached most effectively, while maintaining the concrete's mechanical properties.

**Keywords:** recycling of waste materials; polyethylene; ground granulated blast furnace slag; mechanical properties

## 1. Introduction

Concrete is the most widely used man-made material on the planet and is used in all construction structures. Types of concrete have continuously increased and improved with the development of technology. However, the extensive use of concrete has led to increased carbon dioxide emissions discharged by the cement industry and serious environmental pollution and ecological problems. Cement manufacturing can be identified as a significant source of $CO_2$ emissions, from production to emissions, making it the largest source of industrial emissions [1]. The cement industry is the second largest source of $CO_2$ emissions, accounting for 27% of $CO_2$ emissions from the industrial sector and 8% of global $CO_2$ emissions [2,3]. It is estimated that by 2050, cement production will increase by approximately 12–22% compared to 2014. Therefore, efforts to mitigate $CO_2$ emissions from concrete production have focused on (i) using alternative fuels and raw materials during cement production, (ii) developing alternative low-carbon binders, and (iii) substituting with raw or mineral materials a part of the clinker during cement production or a part of cement during concrete manufacturing [4].

During the process of making iron with blast furnaces in consistently operated steelworks, each ton of steel generates approximately 300 kg of blast furnace slag [5]. Replacing cement with by-products from the ironmaking industry (such as blast furnace slag) can not only effectively reduce cement consumption and carbon emissions, but also achieve the goals of energy conservation and carbon reduction [6]. The hardened blast furnace slag is formed after rapid cooling among the different types of blast furnace slag cooling [7].

Ground Granulated Blast Furnace Slag (GGBFS) belongs to a type of blast furnace slag. GGBFS has become the main construction material to replace cement and can improve the strength of concrete in the late period. GGBFS is widely used in many important engineering projects around the world [8,9]. The GGBFSS generated by regular steelmaking by Sino Steel Corp. has become a resource under long-term study and vigorous promotion in Taiwan.

The GGBFSS is generated by regular steelmaking by Sino Steel Corp of Taiwan in this study. The chemical composition of GGBFS is very similar to that of Portland cement; its composition consists of varying proportions of lime and alumina. Magnesium, silicon, calcium, aluminum, and oxygen make up 95% of the total GGBFS content [10]. Proper use of ground-granulated blast-furnace slag (GGBFS) to replace cement can reduce cement use and slag emissions, and the properties of ground-granulated blast-furnace slag can also be used to improve its engineering properties.

In the 1950s, plastics or synthetic organic polymers were mass-produced and used. Packaging is the largest market for plastic use, and the use of single-use containers is falling. In high and middle-income countries, the proportion of solid waste generated from these containers increased from less than 1% in 1960 to more than 10% in 2005 [11]. Although the growth of plastic use does not yet exceed that of steel and cement materials widely used in civil construction, the impact of plastic waste on the environment and how to deal with and eliminate it is still an important issue that cannot be ignored [12].

Most monomers used to make plastics, such as ethylene and propylene, come from fossil hydrocarbons. None of the common plastics is biodegradable [13]. The only way to permanently eliminate plastic waste is through destructive thermal treatment, such as combustion or pyrolysis, which tends to cause secondary environmental pollution as a result of subsequent carbon dioxide emissions. If the waste is buried in a landfill, it will accumulate only in the natural environment and will not decompose [13,14]. Therefore, plastic waste's almost permanent pollution of the natural environment is a growing problem [15]

Taiwan has the second highest density of convenience stores in the world [16], and most disposable food packages usually contain a plastic film [5] composed of polyethylene (PE), a high-quality material with resistance to impact, good chemical stability and resistance to low temperatures. Most single-use paper containers are considered plastic waste rather than waste paper. With the ever-increasing amount of plastic waste, recycling is an efficient way to reuse or recycle waste into useful products, materials, or components

According to the Taiwan Environmental Protection Agency, in Taipei alone, takeout packaging increased by 85% in May 2021, which may be due to the impact of the COVID-19 outbreak. In general, recycled food packaging consists of different polymers and complex materials, making it difficult to separate each material [16]. The PE components in the packaging container must be separated by buoyancy separation and then recycled.

Polyethylene (PE) is one of the most widely used polymers in the world [17]. One million plastic bottles are purchased every minute and this number is expected to increase by 20% by 2021 [18]. Due to the excessive use of this material, the problems caused by its waste are becoming more and more serious.

In search of sustainable solutions for PE waste, research has been developed using waste as a constituent of concrete, mortar, and fly ash geopolymer in recent decades [19–26]. Based on the concepts of sustainable construction and the past results obtained in the research analyzed. The use of PE waste as a substitute in natural mortar sand has become an interesting option to reconcile the reduction in the use of natural aggregates and the reuse of non-biodegradable waste [4]. Researchers have studied the inclusion of recycled plastic waste in building materials such as bricks and concrete [27,28]. In addition, recycled plastic waste can also be used in plastic-wood building materials when combined with wood and other plant fibers [5]. In recent years, ordinary-strength concrete has not been able to meet performance demands for the increasing number of underground structures in complex geology [29]. This results in an improved global structural response of the

composite, characterized by an increase in different mechanical properties, such as tensile and flexural strengths, ductility, toughness, and fracture energy [30].

In this study, adding GGBFS and waste PE materials to make cement mortar improves the usability of waste and green energy. Research indicates that PE fiber-reinforced CAC-GGBFS blended mortar provides improved durability against sulfuric acid attacks, adding longevity and decreasing the maintenance costs of concrete sewage pipes [31,32].

Past studies have shown that recycled plastics can replace fine aggregate and add mortar [33]. However, as the discarded PE in the mix increases, the bulk density of the mortar in the hardened state decreases. This reduction is due to the fact that the density of plastic aggregate is 70% lower than that of natural sand [34].

The water absorption of the mortar increases with the content of waste PE because the addition of waste PE induces the balling phenomenon, which increases the internal porosity of the specimen, and the water absorption rate increases with the content of the PE material. When 20% sand was replaced by low-density polyethylene (PE), the overall density of the mortar decreased by 10% and the water absorption increased by 11.5% [35].

The measured value of the velocity of the ultrasonic wave has a considerable relationship with the compactness of the cement products. Safi B et al. observed a reduction in the speed of the ultrasonic pulse in mortar samples due to the increase in the aggregate of waste PE in mortars [36]. Da Silva et al. [37] found a reduction in the modulus of elasticity due to the increase in waste PE aggregate in mortars, which is a consequence of the reduction in speed in the ultrasonic pulse.

The compressive strength was reduced due to the increase in the PE aggregate in the mixtures; a strength reduction between 29% and 69% of the reference sample was found for a 50% replacement rate of sand with PE aggregate [38]. Mixtures containing 50–60% low-density polyethylene (LDPE) were found to meet the strength criteria for masonry mortars [35].

The flexural strength is reduced as a function of an increase in the rate of PE aggregate in the mixture. A reduction in flexural strength was observed between 18% and 40% with the 50% replacement rate of sand by waste PE aggregate. Due to the presence of plastic in the mixture, increasing the deformation index of the mortar, the resulting material can absorb shock and stretch, thus becoming more ductile during mortar cracking, thus reducing cracking [34].

PE fiber provides improved durability of mortar to sulfuric acid attacks. Using ground-granulated blast-furnace slag (GGBFS) to replace cement can improve the strength of the concrete in the late period and reduce cement use and slag emissions. We expect to study the ratio of mixing these two materials to improve the engineering properties and durability of the cement mortar. The purpose of this study is to compare research trials with the previous literature using lightweight polyethylene (PE) aggregate added and ground granulated blast furnace slag (GGBFS) to replace cement in cement mortar and to learn about their performance and durability.

## 2. Materials and Methods

### 2.1. Materials Used for the Experiment

The materials used in this study are presented in Figure 1, and their physical properties and chemical compositions are shown in Tables 1 and 2.

1.  The Portland Type I cement produced by the Taiwan Cement Corporation was used, and the properties of this cement conformed to ASTM C150; the specific gravity was 3.15, and the fineness was 3450 $cm^2$/g.
2.  Waste polyethylene (waste PE) in its original shape was a large spherical particle. After being broken down by the grinder, its shape was plastic cotton fiber, as shown in Figure 1b, with a specific gravity of 0.923 and a moisture content of 8.2%. The chemical composition of PE is $(C_{10}H_8O)n$. The Fourier transform infrared spectroscopy (FTIR) spectrum of waste PE is shown in Figure 2.

3.  Ground granulated blast furnace slag (GGBFS) was obtained from the CHC Resources Corporation, and the properties conformed to CNS12549, with a specific gravity of 2.9 and a fineness of 4000 cm$^2$/g.
4.  The fine aggregate was river sand from the Ligang River, and the specific gravity was tested according to ASTM C127. The specific gravity was 2.65, and the water absorption was 1.48%.

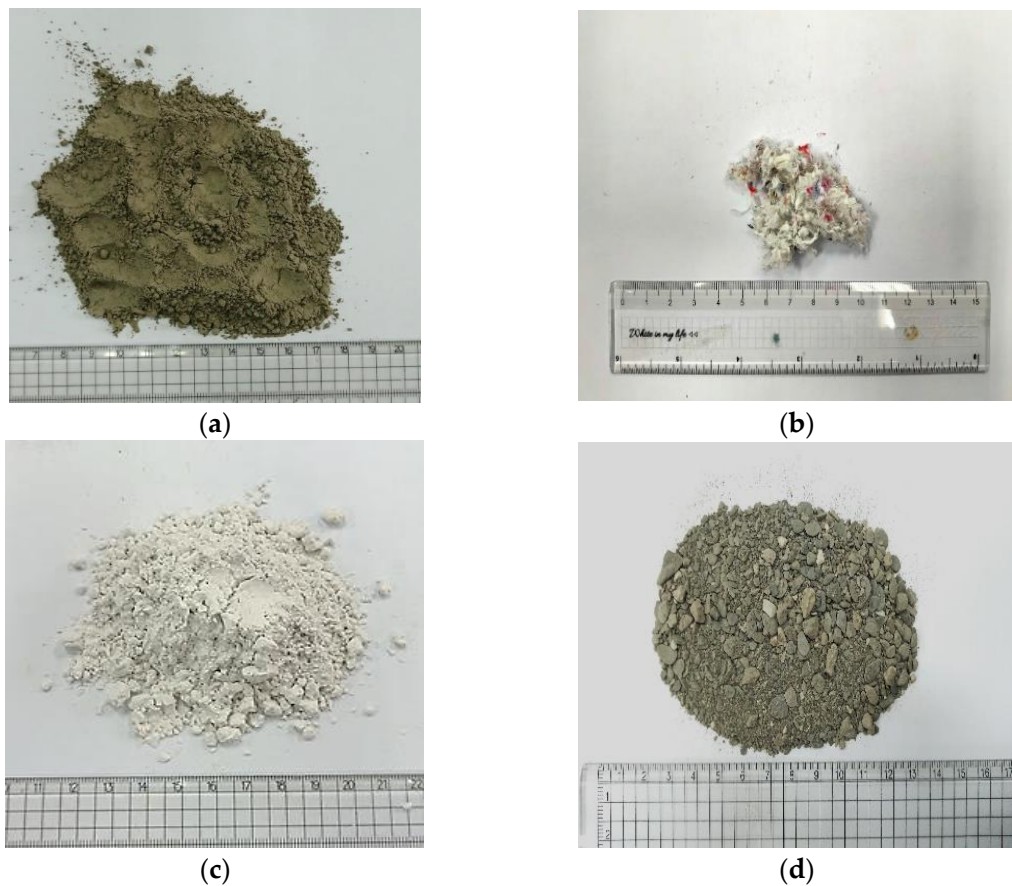

**Figure 1.** Test materials: (**a**) cement, (**b**) waste PE, (**c**) GGBFS, and (**d**) sand.

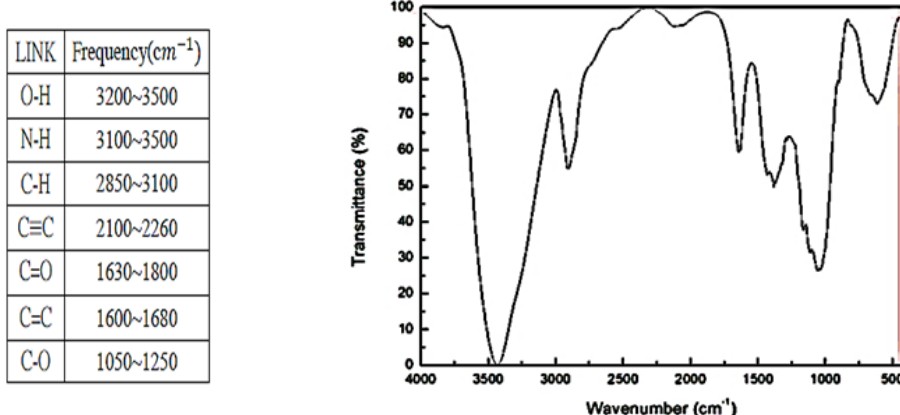

**Figure 2.** Fourier transform infrared spectroscopy (FTIR) analysis of polyethylene.

**Table 1.** Physical properties and chemical composition of test materials.

| Materials | Cement | GGBFS | PE |
|---|---|---|---|
| Physical properties | | | |
| Specific gravity | 3.15 | 2.90 | 0.92 |
| Fineness ($cm^2/g$) | 3450 | 4000 | – |
| Moisture content (%) | – | – | 8.2 |
| Chemical contents (%) | | | |
| $SiO_2$ | 19.6 | 33.5 | |
| $Al_2O_3$ | 4.4 | 14.7 | |
| $Fe_2O_3$ | 3.1 | 0.4 | |
| CaO | 62.5 | 41.2 | |
| MgO | 4.9 | 6.4 | |
| $SO_3$ | 2.2 | 0.6 | |
| $K_2O$ | – | 0.3 | |
| $Na_2O$ | – | 0.2 | |
| $TiO_2$ | 0.5 | 0.5 | |
| $P_2O_5$ | 0.11 | 0.01 | |
| f-CaO | 0.7 | – | |
| $C_3S$ | 56 | – | |
| $C_2S$ | 14 | – | |
| $C_3A$ | 7 | – | |
| $C_4AF$ | 9 | – | |
| L.O.I | 2.5 | 0.58 | |

**Table 2.** Mixture proportions of cement mortars (unit: $kg/m^3$).

| W/B | GGBFS (%) | GGBFS | Cement | PE (%) | PE | Sand | Water |
|---|---|---|---|---|---|---|---|
| 0.5 | 0 | 0 | 822 | 0 | 0 | 2302 | 411 |
| | | | | 1 | 13.8 | | |
| | | | | 2 | 27.7 | | |
| | | | | 3 | 41.5 | | |
| | | | | 4 | 55.4 | | |
| | 10 | 75.7 | 740 | 0 | 0 | | |
| | | | | 1 | 13.8 | | |
| | | | | 2 | 27.7 | | |
| | | | | 3 | 41.5 | | |
| | | | | 4 | 55.4 | | |
| | 20 | 151.3 | 658 | 0 | 0 | | |
| | | | | 1 | 13.8 | | |
| | | | | 2 | 27.7 | | |
| | | | | 3 | 41.5 | | |
| | | | | 4 | 55.4 | | |

### 2.2. Test Specifications and Material Mix Proportions

The fresh properties of cement mortar included slump and flow. For maintenance, the cement mortar samples were taken and placed in saturated lime water. In addition, the engineering properties and durability characteristics were studied at ages 3, 7, 28, 56, and 91 days. Due to the difference in density and volume of waste aggregate and natural aggregate, using a weight replacement ratio is unsuitable for concrete mixes in which waste aggregate and natural aggregate are mixed. Weight substitution does not fit the concept of

packing density. Therefore, we replaced all waste materials added in this study by volume replacement rate instead of weight replacement rate and calculated their equivalent weight. The equivalent weights of all materials are listed in Table 1. The unit weight of the material mix ratio is shown in Table 2. The test methods and specifications of this study are shown in Table 3.

**Table 3.** The test methods and specifications.

| Test Items | | Test Specifications |
| --- | --- | --- |
| 1. | Slump | ASTM C143 |
| 2. | Flow | ASTM C230 |
| 3. | Setting time | ASTM C403 |
| 4. | Compressive strength | ASTM C109 |
| 5. | Flexural strength | ASTM C348 |
| 6. | Tensile strength | ASTM C190 |
| 7. | Water absorption rate | ASTM C1585 |
| 8. | Ultrasonic velocity | ASTM C597 |
| 9. | Resistivity | ASTM C876 |
| 10. | Resistance to sulfate attack | ASTM C1012 |

## 3. Results and Discussions

### 3.1. Slump

As shown in Table 4 and Figure 3, with GGBFS replacing 20% of the volume, the slumps of cement mortar with different proportions of added waste PE (0%, 1%, 2%, 3%, and 4%) were 2.8 cm, 2.6 cm, 2.5 cm, 2.3 cm, and 2.1 cm. Since the waste PE had a high water-containing ability and could accommodate the free water of mortar in PE pores, the slump decreased when the added volume of waste PE increased. The slumps were 2.2 cm, 2.5 cm, and 2.8 cm when the GGBFS replaced cement proportions were 0%, 10%, and 20%, respectively. As seen, with the increase in the replaced volume of GGBFS, the slump of cement mortar increased. Due to the slower hydration effect of GGBFS in the early stage, the initial setting time of mortar was longer. The free water of the cement mortar increased, resulting in the slump of the mortar increasing. When the replaced volume of GGBFS was higher, the mortar's workability improved. In conclusion, the increase in waste PE addition resulted in a decrease in the mortar slump; the mortar workability increased with the increase in the amount of GGBFS replacement.

### 3.2. Flow

As shown in Table 3 and Figure 4, the flows of cement mortar with different added volumes of waste PE were 19.9 cm, 19.7 cm, 19.4 cm, 18.9 cm, and 18.7 cm. When the added volume was increased from 1% to 4%, the flow decreased by 1.01% to 6.03%, which indicated that the flow decreased as the added volume of waste PE increased. When the W/B was 0.5, 2% waste PE was added with 0%, 10%, and 20% GGBFS to replace cement, and the flows were 17.8 cm, 18.2 cm, and 19.4 cm, respectively. As demonstrated, with the increase in the replaced volume of GGBFS, the flow of cement mortar increased. Due to GGBFS's fineness and its particle shape, a lower relative density increases workability. When the replaced volume of GGBFS increased, the flow also increased.

**Table 4.** Fresh properties of cement mortar with the addition of different ratios of waste PE and GGBFS replacement.

| GGBFS RM (%) | PE AM (%) | Slump (cm) | Flow (cm) | Initial Setting (min) | Final Setting (min) |
|---|---|---|---|---|---|
| 0 | 0 | 2.2 | 18.2 | 317 | 428 |
| | 1 | 1.9 | 17.9 | 304 | 416 |
| | 2 | 1.8 | 17.8 | 291 | 407 |
| | 3 | 1.6 | 17.7 | 278 | 392 |
| | 4 | 1.5 | 17.4 | 269 | 379 |
| 10 | 0 | 2.5 | 18.9 | 368 | 496 |
| | 1 | 2.4 | 18.6 | 360 | 488 |
| | 2 | 2.2 | 18.2 | 346 | 473 |
| | 3 | 2 | 17.8 | 333 | 462 |
| | 4 | 1.9 | 17.4 | 318 | 447 |
| 20 | 0 | 2.8 | 19.9 | 401 | 518 |
| | 1 | 2.6 | 19.7 | 387 | 505 |
| | 2 | 2.5 | 19.4 | 375 | 496 |
| | 3 | 2.3 | 18.9 | 367 | 488 |
| | 4 | 2.1 | 18.7 | 356 | 474 |

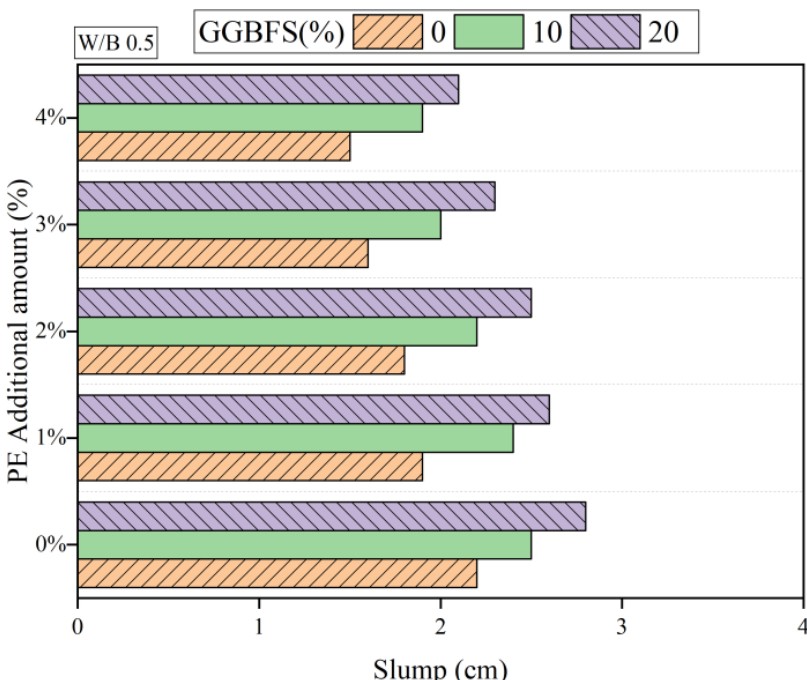

**Figure 3.** The slump of cement mortars with the addition of different ratios of waste PE and GGBFS replacement.

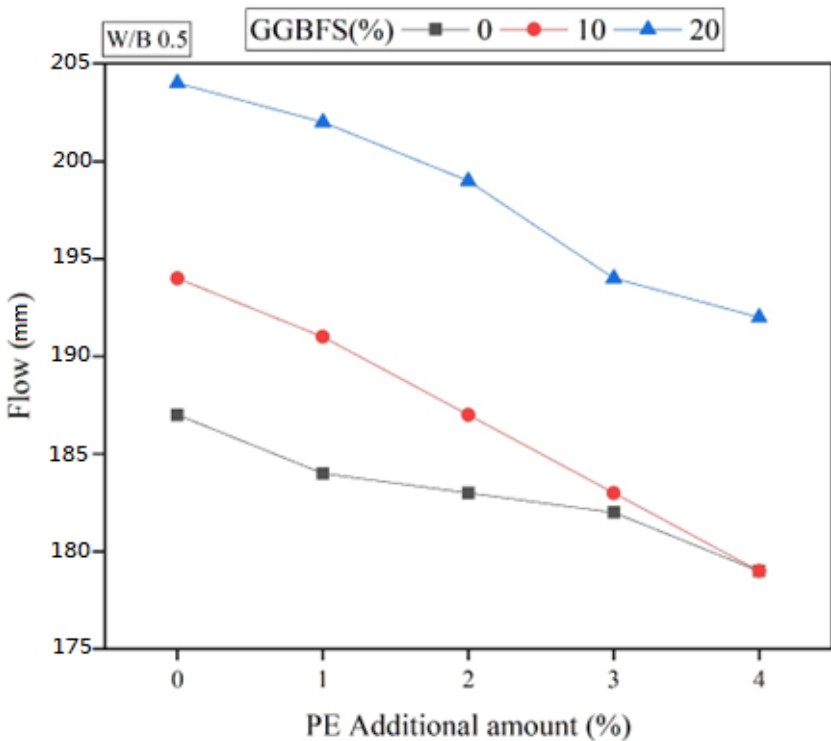

**Figure 4.** The flow of cement mortars with the addition of different ratios of waste PE and GGBFS replacement.

### 3.3. Setting Time

As shown in Table 4 and Figure 5, when the W/B was 0.5, the cement mortar's initial and final setting times with 10% added waste PE and GGBFS were 368 min to 318 min and 496 min to 447 min, respectively. This indicated that the setting time shortened as the added waste PE increased. Since the texture of waste PE could accommodate the free water of the mortar, the water absorption amount of mortar grew as the volume of waste PE was added, reducing the free water of the mortar and accelerating the reaction of the cement's hydrothermal chemistry, leading to a short setting time of the mortar. If we added 2% waste PE with 0%, 10%, and 20% GGBFS to replace the cement, the initial setting times were 291 min, 346 min, and 375 min, respectively, and the final setting times were 407 min, 473 min, and 496 min, respectively. It shows that with the increase in the replaced volume of GGBFS, the setting time of the cement mortar increased. The GGBFS as a cement replacement had lesser water demand because of its glassy texture, and the glassy surface of the GGBFS particles did not absorb water onto its surface. At a fixed water-to-binder ratio, the volume of the floating water increase led to a prolonged setting time. The reduced total volume of the cement was due to the replacement of the cement by GGBFS, and the heat of hydration decreased. Therefore, the hydration reaction activity of the mortar declined, which delayed the setting time.

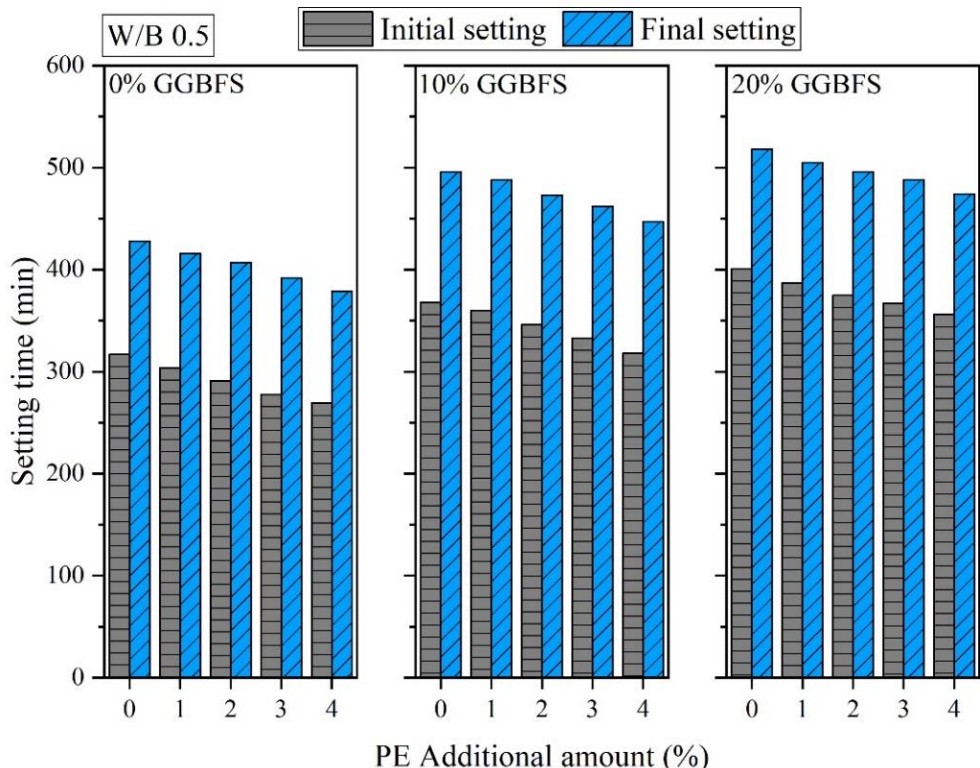

**Figure 5.** Setting time of cement mortars with the addition of different ratios of waste PE and GGBFS replacement.

Overall, with the added waste PE, the setting time showed a shortened trend, while the cement replacement by GGBFS resulted in a lengthened setting time.

### 3.4. Compressive Strength

As shown in Table 5 and Figure 6, when the proportion of GGBFS was 0% on the third day, the compressive strength of the control group was 29.6 MPa, and the compressive forces of groups with 1% to 4% waste PE were 28.3 MPa to 25 MPa (decreased by 5.5% to 16.9%, respectively). After 28 days, the strength of the control group with 0% GGBFS was 45.2 MPa; the compressive forces of groups with 1% to 4% waste PE were 42.8 MPa to 41.6 MPa (decreased by 5.4% to 8.1%). On the 91st day, the strength of the control group was 53.3 MPa; the compressive forces of groups with 1% to 4% waste PE were 51.4 MPa to 48.2 MPa (decreased by 3.6% to 9.5%). We observed that as the waste PE increased, the compressive strength decreased. Due to waste PE having water absorbability and absorbing free water from the mortar, it led to an increased number of holes in the mortar.

The 28-day compressive strength decreased with the PE aggregate content in the mixtures. The drop in compressive strength could be attributed to the poor bonding between the matrix and plastic aggregates [10].

**Table 5.** Compressive strength of cement mortars with the addition of different ratios of waste PE and GGBFS replacement (unit: MPa).

| W/B | GGBFS (%) | PE (%) | 3 Days | 7 Days | 28 Days | 56 Days | 91 Days |
|---|---|---|---|---|---|---|---|
| | | 0 | 29.6 | 35.7 | 45.2 | 49.1 | 53.3 |
| | | 1 | 28.3 | 33.7 | 42.8 | 47.6 | 51.4 |
| | 0 | 2 | 27.3 | 33.8 | 43.3 | 47.7 | 51.1 |
| | | 3 | 26.5 | 33.0 | 42.9 | 47.1 | 49.8 |
| | | 4 | 24.9 | 32.5 | 41.6 | 46.2 | 48.2 |
| | | 0 | 27.1 | 34.4 | 46.3 | 51.2 | 56.0 |
| | | 1 | 25.4 | 32.8 | 45.6 | 49.3 | 55.1 |
| 0.5 | 10 | 2 | 24.3 | 32.1 | 44.9 | 48.8 | 53.9 |
| | | 3 | 23.6 | 30.6 | 43.3 | 47.8 | 53.4 |
| | | 4 | 22.8 | 30.3 | 42.0 | 45.8 | 51.9 |
| | | 0 | 24.7 | 33.3 | 48.5 | 53.5 | 59.1 |
| | | 1 | 23.0 | 33.1 | 47.8 | 52.5 | 57.9 |
| | 20 | 2 | 21.6 | 30.0 | 46.8 | 50.0 | 55.5 |
| | | 3 | 21.0 | 29.4 | 45.9 | 51.1 | 56.9 |
| | | 4 | 20.3 | 28.6 | 44.9 | 49.3 | 53.8 |

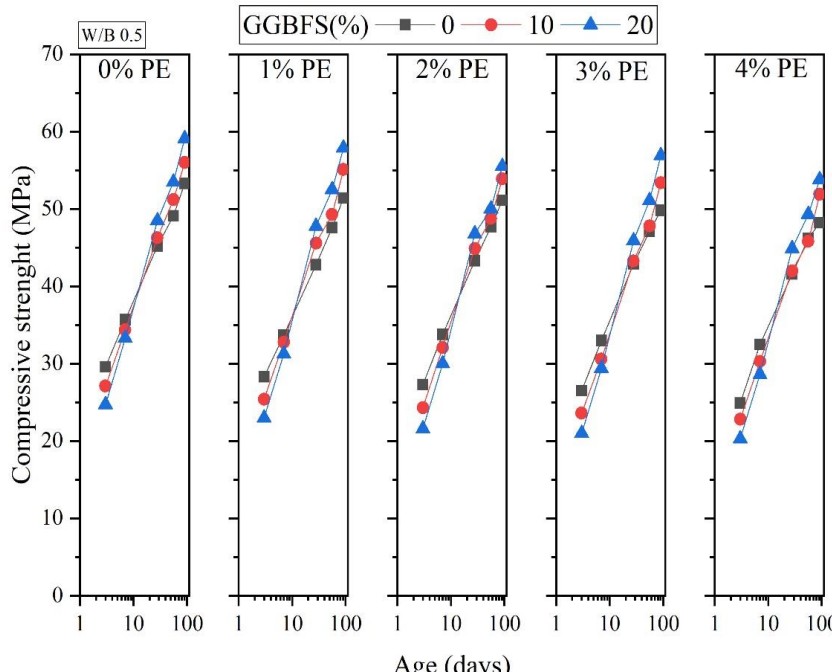

**Figure 6.** Compressive strength of cement mortars with the addition of different ratios of waste PE and GGBFS replacement.

After 28 days, the proportion of the replaced volume of GGBFS was 10%, and the compressive strength of the groups decreased by 1.5% to 3.0% when the added waste PE increased from 1% to 2%. The volume of the added waste PE rose from 3% to 4%, and the compressive strength of the groups was reduced by 6.5% to 9.3%. After 56 days, the volume of the added waste PE increased from 1% to 2%, and the compressive strength of the groups decreased by 3.7% to 4.6%. The added waste PE rose from 3% to 4%, and the compressive

strength of groups decreased by 6.6% to 10.5%. When the added waste PE was 1% to 2%, the decrease in strength was controlled to fall within 5%, which could achieve the function of removing wastes, reducing waste generation, and improving the economic efficiency of wastes for sustainable development of resources on Earth. After 91 days, the compressive strength of the control group was 53.3 MPa to 48.2 MPa, while that of the group with 10% added GGBFS was 56 MPa to 51.9 MPa (increased by 5.1% to 7.6%, respectively), and that of the group with 20% added GGBFS was 59.1 MPa to 53.8 MPa (increased by 10.8% to 11.5%, respectively). Since the added GGBFS produced pozzolanic reactions constantly during the late period, the strength of the cement mortar further improved.

After 28 days, the proportion of the replaced volume of GGBFS was 10%, and the compressive strength of the groups decreased by 1.5% to 3.0% when the added waste PE increased from 1% to 2%. The strength of the groups was reduced by 6.5% to 9.3% when the added waste PE increased from 3% to 4%.

After 56 days, the compressive strength of the groups decreased by 3.7% to 4.6% with the volume of 1% added PE to 2% to mortar, while that of the groups with 3% added PE to 4% decreased by 6.6% to 10.5%.

### 3.5. Flexural Strength

As shown in Figure 7, on the 3 days with the W/B at 0.5, the flexural strength without adding GGBFS and waste PE was 9.1 MPa to 5.9 MPa, which indicated that the flexural strength lowered with increasing added volume (7.7% to 35.2%). After 28 days, the flexural strength of the control group was 15.8 MPa, the added volume of waste PE increased to 4%, and the flexural strength was 12.7 MPa (decreased by 19.6%). On the 91st day, the flexural strength of the control group was 22.7 MPa. The flexural strength (19.9 MPa) was the lowest when the added waste PE was 4%. When the added volume of waste PE increased from 1% to 4%, the flexural strength decreased by 2.3% to 15.8%.

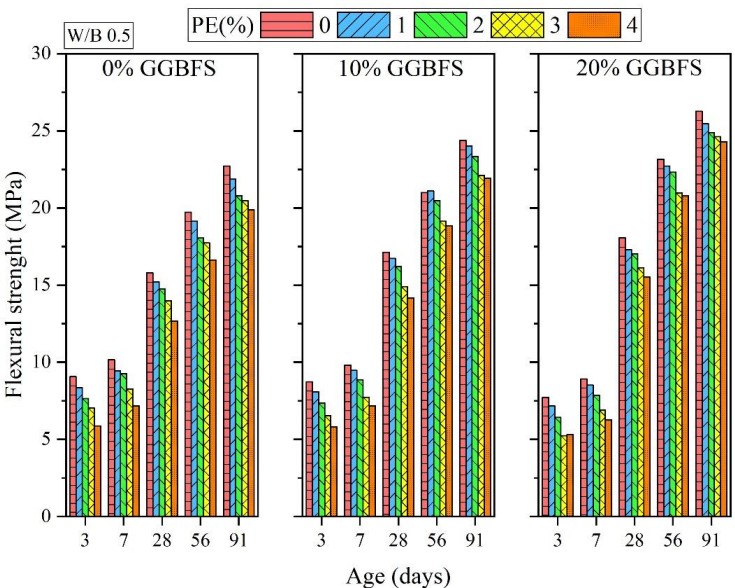

**Figure 7.** Flexural strength of cement mortars with the addition of different ratios of waste PE and GGBFS replacement.

After 56 days, the replaced volume of GGBFS was 10%, added waste PE was 1% to 2%, and the flexural strength decreased by 0.5% to 2.4%. When the added waste PE was 3% to 4%, the flexural strength dropped by 8.6% to 10%. After 91 days, the added waste PE was 1% to 2%, and the flexural strength dropped by 1.6% to 4.5%. When the added waste PE was 3% to 4%, the flexural strength dropped by 9.4% to 10.2%. This result indicated that GGBFS could continue the pozzolanic reaction during the late period and increase flexural strength.

The experimental results showed that when the waste PE content was less than 2%, the percentage of strength reduction was within 5%. This way, the waste could be effectively recycled, and energy consumption and environmental pollution could be reduced. As the replaced volume of GGBFS increased, the increase in strength in the early period was slow, but that in the late period was obvious under the pozzolanic reaction. The flexural strength could be the highest when GGBFS was 20% in the late period, followed by the flexural strength when GGBFS was 10%. To sum up the above, the addition ratio of waste PE should be controlled within 2%, which will effectively reuse waste and reduce pollution.

### 3.6. Tensile Strength

As shown in Figure 8, on days 3, 28, and 91 with the W/B at 0.5, the tensile strengths with different proportions of waste PE were 6.7 MPa to 5.1 MPa, 10.2 MPa to 8 MPa, and 14 MPa to 11.8 MPa. On the 3 days without adding GGBFS, the added volume of waste PE increased from 0% to 4%, and the tensile strength decreased by 2.1% to 15.7%. This result indicated that the tensile strength had a slightly decreasing trend as the waste PE increased. After 91 days, only 1% of the waste PE was added, and the tensile strength was 13.7 MPa. When 1% of the waste PE was added together with 10% and 20% GGBFS, the tensile strengths were 14.6 MPa and 15.7 MPa, respectively.

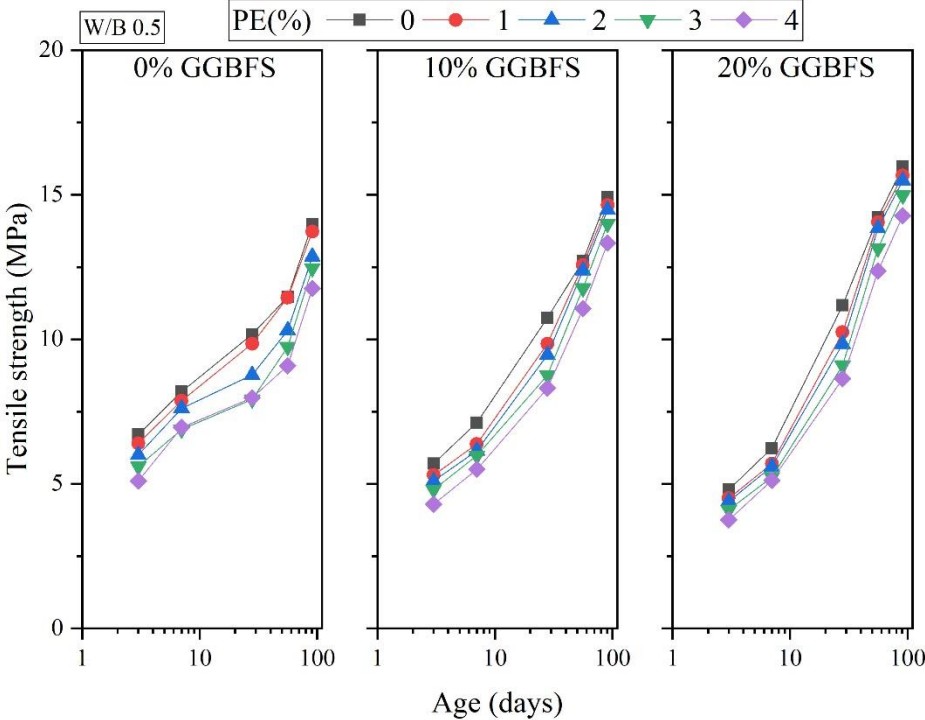

**Figure 8.** Tensile strength of cement mortars with the addition of different ratios of waste PE and GGBFS replacement.

From days 28 to 91, after adding 1% waste PE, the tensile strength increased by 38.4%. When 1% waste PE was added together with 10% and 20% GGBFS, the tensile strength was enhanced by 46.0% and 53.9%, respectively. This result indicated that the added GGBFS continued the pozzolanic reaction in the late period, which further enhanced the tensile strength of the cement mortar, and the compressive and flexural strength also showed the same trend.

### 3.7. Ultrasonic Velocity

The ultrasonic velocity measurement method will not cause any damage to the cement products, and the simple operation is a method that the engineering field is willing to

adopt. The measured wave velocity value is closely related to the compactness of the cement products. Generally speaking, the pulse wave velocity is proportional to the square root of the elastic modulus of concrete and inversely proportional to the square root of the concrete density. The elastic modulus of concrete is proportional to the square root of the compressive strength according to the ACI 318 recommendation, and the pulse wave velocity is proportional to the fourth root of the compressive strength. That means that when the compressive strength of concrete increases with the increase in the material's age, the pulse wave velocity will also increase slightly. However, at a later curing period of the material, the pulse wave velocity in the concrete does not change sensitively with the strength.

After 3 days, the W/B was 0.5, and the ultrasonic velocities varied from 3862 m/s to 3589 m/s with different proportions of waste PE, as shown in Table 6 and Figure 9. After 28 days, the wave velocities of the control group and other groups with different proportions of GGBFS (0%, 10%, and 20%) replacing cement were 4192 m/s, 4300 m/s, and 4386 m/s (increased by 2.6% to 4.5%). This result indicated that the replaced volume of GGBFS increased from 0% to 20%, and the ultrasonic velocity tended to rise slightly due to the pozzolanic reaction of GGBFS. After 91 days, the ultrasonic velocities of the control group and other groups with 1% to 4% added waste PE were 4616 m/s, 4554 m/s, and 4310 m/s, while the ultrasonic velocity decreased by 1.3% and 6.6% compared with the control group. This result indicated that the ultrasonic velocity decreased due to the increment in the waste PE.

**Table 6.** Ultrasonic pulse velocity of cement mortars with the addition of different ratios of waste PE and GGBFS replacement (unit: m/s).

| W/B | GGBFS (%) | PE (%) | 3 Days | 7 Days | 28 Days | 56 Days | 91 Days |
|---|---|---|---|---|---|---|---|
| 0.5 | 0 | 0 | 3862 | 3975 | 4192 | 4391 | 4616 |
| | | 1 | 3830 | 3973 | 4137 | 4302 | 4554 |
| | | 2 | 3800 | 3911 | 4083 | 4256 | 4462 |
| | | 3 | 3672 | 3880 | 4080 | 4223 | 4395 |
| | | 4 | 3589 | 3799 | 3970 | 4174 | 4310 |
| | 10 | 0 | 3785 | 3816 | 4300 | 4523 | 4801 |
| | | 1 | 3743 | 3783 | 4261 | 4430 | 4720 |
| | | 2 | 3689 | 3764 | 4225 | 4384 | 4667 |
| | | 3 | 3573 | 3725 | 4189 | 4361 | 4572 |
| | | 4 | 3507 | 3674 | 4116 | 4300 | 4482 |
| | 20 | 0 | 3586 | 3750 | 4386 | 4650 | 4903 |
| | | 1 | 3538 | 3707 | 4332 | 4561 | 4816 |
| | | 2 | 3489 | 3681 | 4257 | 4513 | 4762 |
| | | 3 | 3422 | 3673 | 4198 | 4492 | 4665 |
| | | 4 | 3400 | 3601 | 4173 | 4427 | 4607 |

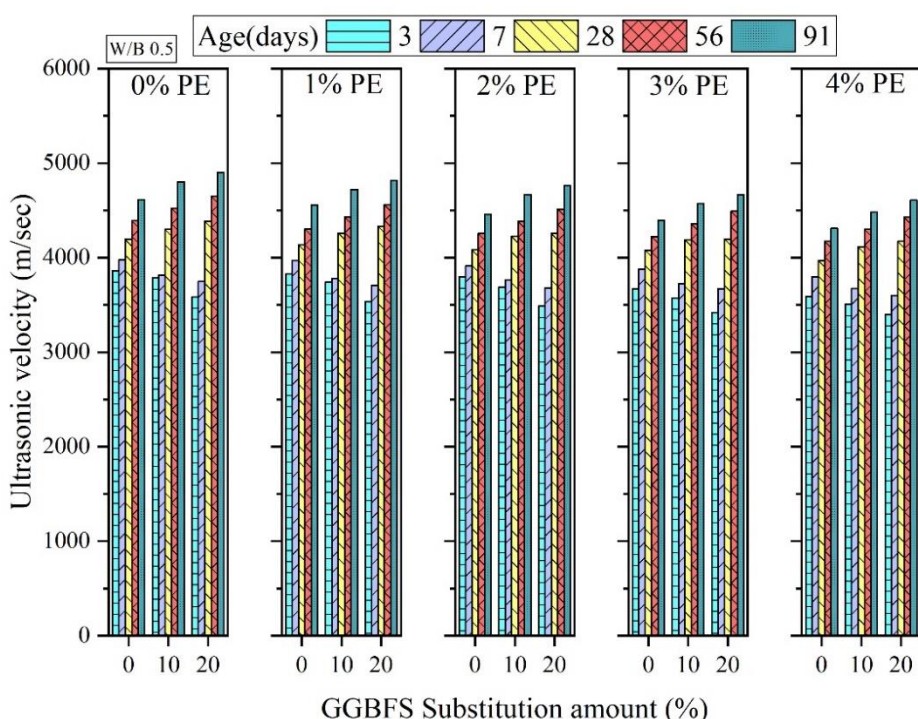

**Figure 9.** The ultrasonic velocity of cement mortars with the addition of different ratios of waste PE and GGBFS replacement.

After 91 days, the wave velocities of the groups without added waste PE but with different proportions of GGBFS (0%, 10%, and 20%) to replace cement were 4616 m/s, 4801 m/s, and 4903 m/s, respectively. From 28 to 91 days, the wave velocity of the control group increased by 10.1%. When the replaced volumes of GGBFS were 10% and 20%, the wave velocities increased by 11.7% and 11.9%, respectively. This result indicated that the groups with GGBFS added produced pozzolanic reactions in the late period, which filled the internal holes of the sample and increased its compactness. After 56 days, when the replaced volume of GGBFS was 20%, and 1% to 2% of waste PE was added, the ultrasonic velocity was higher than 4500 m/s, which indicated that the concrete was of good quality.

*3.8. Water Absorption Ratio*

As shown in Figure 10, when the W/B was 0.5 on the 28th day, with no added GGBFS and 0%, 1%, 2%, 3%, and 4% waste PE added to the cement mortar sample, the water absorptions were 10%, 10.3%, 10.6%, 11.1%, and 11.6%, respectively. This result indicated that when the added volume of waste PE increased, the sample generated more holes, which increased the internal holes of the sample and water absorption.

After 28 days, when adding 10% GGBFS to replace the cement, the water absorption was 10.2% to 10.9%; when 20% GGBFS was added, the water absorption was 8.7% to 10.5%. If the replaced volume of GGBFS increased, hydration in the early phase was slow, which made the water absorption of the sample higher than that of the control group. However, the hydrates produced by the pozzolanic reaction of GGBFS filled the pores of the samples and made the mortar structure dense. In the late period, when the replaced volume of GGBFS was higher, the decline in water absorption was more prominent. This result indicated that GGBFS could increase the sample's internal compactness and decrease permeability, and the durability could be higher. The group with 20% GGBFS had lower water absorption, good compactness, and higher durability.

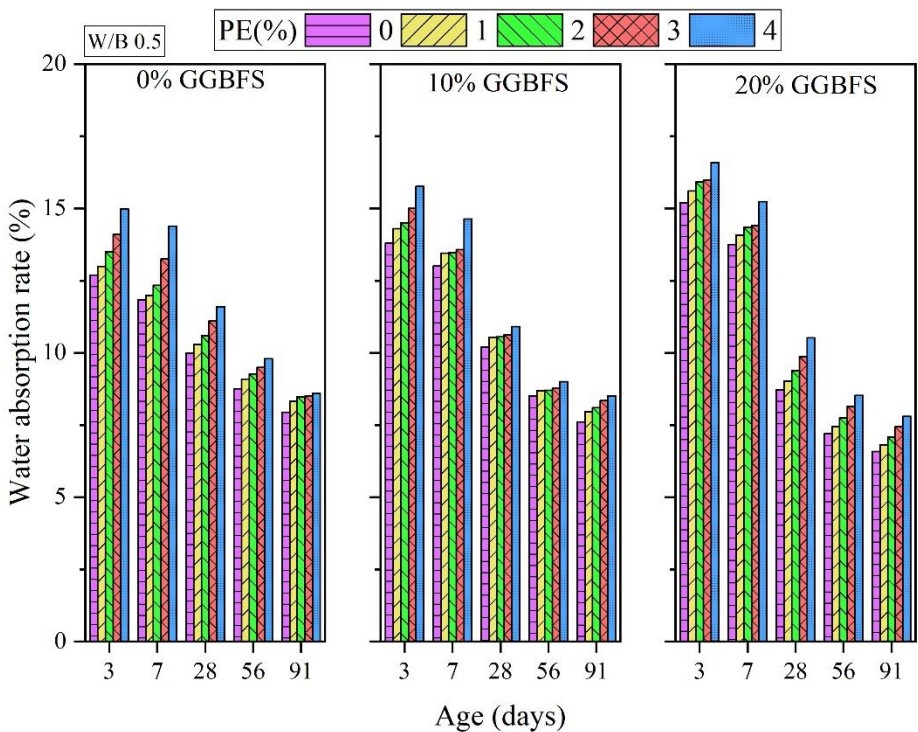

**Figure 10.** The water absorption rate of cement mortars with the addition of different ratios of waste PE and GGBFS replacement.

*3.9. Resistivity*

At present, the resistance value and electro-osmotic value of concrete are used to evaluate its durability. The resistance value measures the conductivity of the concrete surface. The electrical conductivity of concrete materials is carried out by the movement of ions caused by electrolytic reactions in the pores. Concrete with more pore water or a high water–cement ratio has a lower resistivity. The fewer the pores of the concrete, the higher the density, the longer the electrical conduction path, and the higher the resistance value. The resistivity of concrete significantly influences the corrosion rate of steel bars. When the resistance value of concrete is less than 10 k2-cm, the probability of corrosion is high. Still, the resistance value of concrete is greater than 50 k2-cm, which can significantly reduce the corrosion rate of steel bars in concrete.

As shown in Figure 11, for the curing age from 3 to 7 days with W/B = 0.5, the volume of waste PE increased from 0 to 4%, and the resistivity decreased from 11.3 kΩ·cm to 9.3 kΩ·cm. After 28 days, the control group had a higher resistivity of 25.1 kΩ·cm. When the added volume of waste PE increased to 4%, the resistivity was 22.9 kΩ·cm (−8.8%), which showed the resistivity declined due to the increase in the added volume of waste PE. After 91 days, the resistivity of the control group was 35.2 kΩ·cm. When adding 4% waste PE, the resistivity was 31.5 kΩ·cm. When the added volume of waste PE rose from 0% to 4%, the resistivity decreased by 10.5%. This result indicated that the sample structure had higher compactness in the late period. After 28 days with the W/B at 0.5, when 1% waste PE was added, together with 0%, 10%, and 20% GGBFS, the resistivities were 24.5 kΩ·cm, 25.0 kΩ·cm, and 25.4 kΩ·cm, respectively (increased by 2.0% to 3.7%).

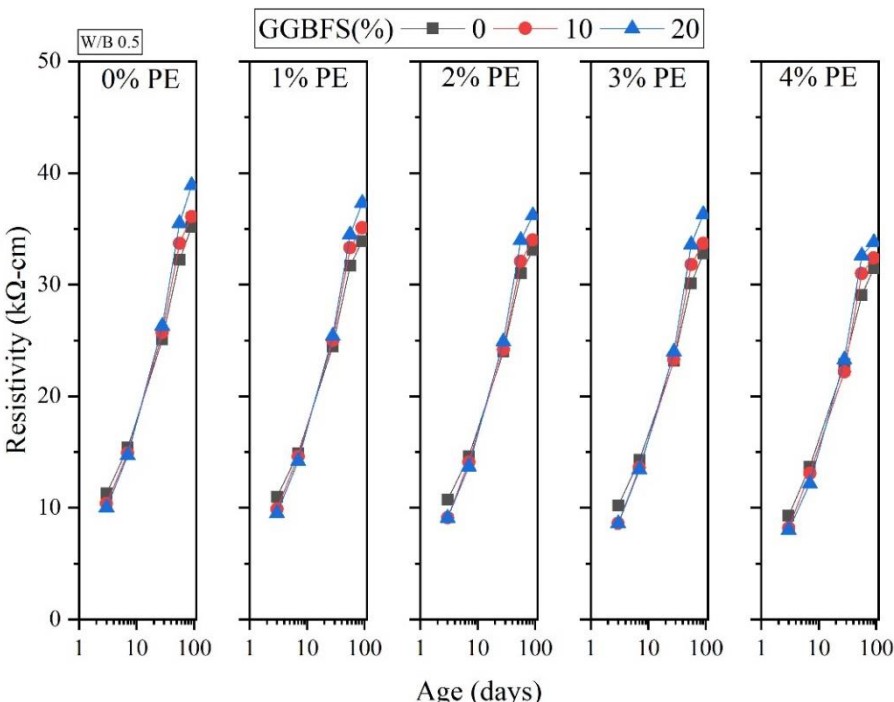

**Figure 11.** The resistivity of cement mortars with the addition of different ratios of waste PE and GGBFS replacement.

This result showed that when the replaced volume of GGBFS increased from 0% to 20%, GGBFS produced a pozzolanic reaction to make the compactness of the internal body of the sample higher than that of the control group. After 91 days, the resistivities were 35.2 kΩ·cm, 36.1 kΩ·cm, and 38.9 kΩ·cm when 1% waste PE was added and when 0%, 10%, and 20% GGBFS replaced the cement, respectively. This result indicated that the groups with different proportions of GGBFS produced pozzolanic reactions in the late period, which made the sample more compact. When 20% GGBFS was added in the late period, the resistivity was higher, followed by the group with 10% GGBFS. After 28 days, the resistivity of all groups with different proportions was higher than 20 kΩ·cm, and these groups all showed durability.

### 3.10. Resistance to Sulfate Attack

As shown in Figure 12, when the W/B was 0.5 after 28 days, the weight loss rate of the sample of cement mortar with different proportions of waste PE added was −7.5% to −8.9% after five recycles. This result indicated that when the added volume of waste PE increased, the weight loss was higher, and the ability to resist sulfate was worse. The reason was mainly the increased waste PE, which led to increased internal holes that allowed the sulfate solution to erode the sample.

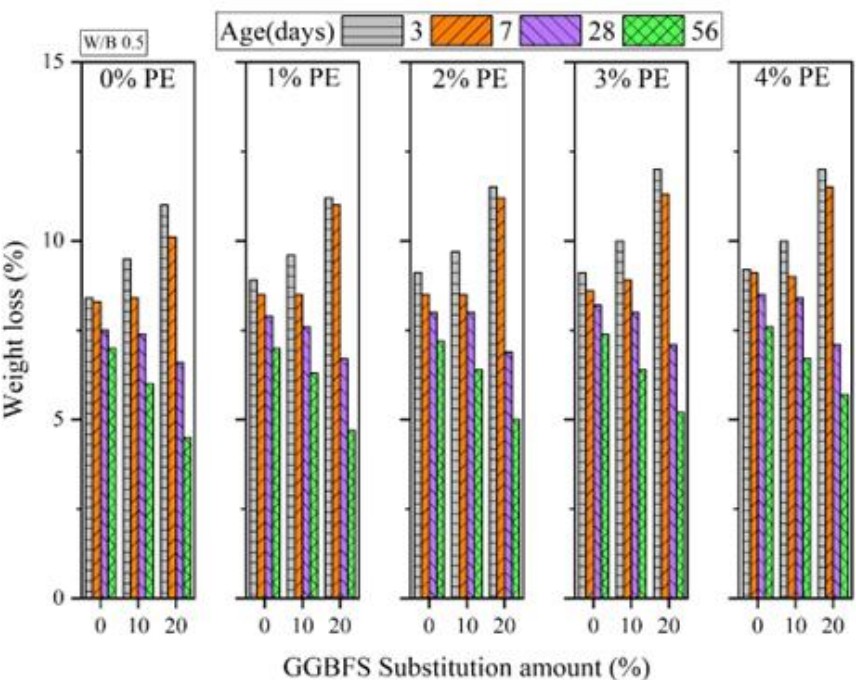

**Figure 12.** Weight loss of cement mortars with the addition of different ratios of waste PE and GGBFS replacement.

After 28 days, the weight loss rate of all proportions of waste PE with the addition of 10% GGBFS increased by −7.4% to −8.4%, while added 20% GGBFS increased by −6.6% to −7.1%, respectively. This result indicated that the increase in the replaced volume of GGBFS could reduce the improvement in the erosion ability of sulfate.

In summary, if the added volume of waste PE increased, it generated more holes. If the replaced volume of GGBFS increased, hydration in the early phase was slow, and the weight loss rate of the sample could increase. However, the hydrates produced by the pozzolanic reaction of GGBFS filled the pores of the samples and made the mortar structure dense. In the late period, the decrease in the weight loss rate was apparent, and when 20% GGBFS was replaced, the durability was higher.

## 4. Conclusions

1. As the added volume of waste PE increased and the replaced volume of GGBFS decreased, the slump and flow declined.
2. The added volume of waste PE increased, and the setting time showed a shortened trend; the replaced volume of GGBFS increased, and the setting time showed a lengthened trend.
3. The compressive, flexural, and tensile strengths of mortar decreased with the increase in the volume of waste PE added. Adding 2% volume of waste PE to the mortar can meet the needs of general engineering use and can achieve the effect of waste recycling. In the later stage of mortar curing, due to the pozzolanic reaction with the addition of GGBFS, with the increase in curing time and the increase in replacement amount, the later-stage strength of mortar increased obviously. When the GGBFS replacement amount was 20%, the strength of the mortar was the best. This amount of replacement increased the overall strength of the specimens.
4. The ultrasonic velocity increased with the curing time. When the addition of waste PE rose from 1% to 4%, the ultrasonic speed decreased by 6.9% to 8.7%. Adding GGBFS can increase the specimen's density in the later stage and then increase the ultrasonic velocity. The ultrasonic speed was the highest at 5172 m/s when using 20% GGBFS instead of cement.

5.  Adding waste PE to the mortar resulted in the generation of holes in the sample and increased the water absorption ratio. Using GGBFS to replace part of the cement, the produced hydrates could fill the holes of the specimen in the late stage of hydration, making the sample denser and lower in water absorption. When we used 20% GGBFS to replace the cement, the water absorption of the mortar reached its lowest (3.7%).

6.  The resistivity decreased as the added volume of waste PE increased. When we gradually increased the replacement of GGBFS, the increase in the mortar resistivity was slow in the early phase, while it increased significantly in the late stage. When the replacement of GGBFS was 20%, we achieved the best result (39 k$\Omega$·cm). After 28 days, the results of all proportions of the mortar were higher than 20 k$\Omega$·cm, reaching the durability requirement for engineering use.

7.  The added volume of waste PE increased, leading to poor resistance to sulfate attack. We used the GGBFS to replace cement, and the hydration of the mortar in the early curing phase was slow and increased the weight loss rate of the sample. The addition of 2% waste PE and the use of 20% GGBFS to replace cement in the range of this study resulted in the sulfate attack resistance of mortar being better.

8.  We recommend adding 2% waste PE and replacing the cement with 10% GGBFS, so the cement products have proper engineering application performance. This is expected to simultaneously have the effects of waste recycling, energy savings, and carbon emission reduction.

**Author Contributions:** Conceptualization, C.-C.H., J.-N.C. and H.-Y.W.; Formal analysis, C.-C.H. and J.-N.C.; Resources, J.-N.C.; Data curation, F.-L.W.; Writing—original draft, C.-C.H. and J.-N.C.; Supervision, H.-Y.W. All authors have read and agreed to the published version of the manuscript.

**Funding:** This research received no external funding.

**Conflicts of Interest:** The authors declare no conflict of interest.

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
