# Peer review of "Effect of Adding Waste Polyethylene and GGBFS on the Engineering Properties of Cement Mortar"

_applsci, doi:10.3390/app122412665_

Round 1

Reviewer 1 Report

Dear Authors,

I read your manuscript carefully. Most of my remarks I inserted into the body text.

You put much effort into the experiment but the rest (English, Abstract, Introduction, way of result presentation, result discussion - too poor without references to literature) dissapoints.

There are too much to change. So, I am opposed to publishing that.

Author Response

Response to Reviewer #1 Comments

applsci-1989274

Title: Effect of adding waste polyethylene and GGBFS on the engineering properties of cement mortar.

applied sciences

Thanks again for your advice and attached is the revised checking list according to your notice.

Point 1:

I read your manuscript carefully. Most of my remarks I inserted into the body text.

You put much effort into the experiment but the rest (English, Abstract, Introduction, way of result presentation, result discussion - too poor without references to literature) disappoints.

There is too much to change.

Response 1:

We will make corrections to the paper and endeavor to make additions and clarifications regarding the English language, grammar, abstract, introduction, presentation of results, discussion of results, and comparison with references.

The following sections provide a list of the important modifications:

Point 2:

Abstract requires total reconstruction.

Response 2:

Thanks for your advice. It was corrected as indicated.

A comprehensively reconstructed abstract focuses on a clear and concise description of the study's objectives, materials, methods, results, and conclusions.

It should be …The recycling of waste materials has become an important topic worldwide. Wastes can be effectively used in concrete to improve its characteristics. This study aimed to research cement mortar's physical, mechanical, and durability. In a cement mortar with a fixed water-to-binder ratio (W/B) of 0.5, waste polyethylene (PE) was added at a sand volume ratio of 0%, 1%, 2%, 3%, and 4%. Replace cement with 0%, 10%, and 20% ground granular blast furnace slag (GGBFS). The results show that the slump and flow of mortar tended to decline as the added amount of waste PE increased, but they also increased with the increased replaced amount of GGBFS. The setting time of mortar was shortened as the increased waste PE but delayed as the increased amount of GGBFS. In terms of mechanical properties, the compressive strength of mortar declined as the replaced amount of waste PE increased. Using the GGBFS to replace part of the cement can improve the later mortar strength. This study found that when the added waste PE is within 2% and the replacement amount of GGBFS is 10%  showed that the goal of recycling waste could reach most effectively and maintain the concrete mechanical properties.

Point 3:

The water absorption of mortar increased as the content of waste plastic increased.

Why? You have explained the reason for density reduction. So, there is my question.

Response 3:

Thanks for your advice. It was corrected as indicated.

The water absorption of the mortar is increased with the waste PE content because the addition of waste PE induces the balling phenomenon, which increases the internal porosity of the specimen, and the water absorption rate increases with PE material content.

Point 4:

Safi B et al. observed a reduction in the ultrasonic pulse speed in the mortar samples due to the waste PE aggregate increase in the mortars [26].

As a factor of what?

Response 4:

Thanks for your advice. It was corrected as indicated.

The measured ultrasonic wave velocity value has a considerable relationship with the compactness of the cement products. Generally speaking, the pulse wave velocity is proportional to the square root of the elastic modulus of concrete and inversely proportional to the square root of the concrete density.

Point 5:

We expect the complementary characteristics of the two materials to improve the various engineering properties and durability of cement mortar.

Write in precise way, it is too general.

Response 5:

Thanks for your advice. It was corrected as indicated.

Research points out that PE fiber provides improved durability of mortar to sulfuric acid attacks adding longevity and decreasing the maintenance costs of concrete sewage pipes. Using ground-granulated blast-furnace slag (GGBFS) to replace cement can enhance the strength of the concrete in the late period, reduce cement use and slag emissions, and improve the engineering properties of cement products. We expect to study the ratio of mixing these two materials to improve the engineering properties and durability of the cement mortar.

Point 6:

The purpose of this study is … and to learn about the sustainability of the developed materials about their performance and durability.

“to learn about the sustainability''. Such approach is not correct and it is too general.

Response 6:

Thanks for your advice. It was corrected as indicated.

It should be…The purpose of this study is to compare research trials with past literature by using the added waste polyethylene (PE) lightweight aggregate and ground-granulated blast furnace slag (GGBFS) to replace cement in cement mortar and to learn about their performance and durability.

Point 7:

  1. Cement:

The Portland Type I cement produced by the Taiwan Cement Corp….

The sentence must be corrected grammatically.

Response 7:

Thanks for your advice. It was corrected as indicated.

After modification, it looks like this:

  1. Cement:

The Portland Type I cement produced by the Taiwan Cement Corporation was used; the properties of this cement conformed to ASTM C150, the specific gravity was 3.15, and the fineness was 3,450 cm2/g.

Point 8:

About 2. Materials and Methods section…

Response 8:

Thanks for your advice. It was corrected as indicated.

It should be …

  1. Materials and Methods

2.1 Materials used for the experiment

Materials used in studies are presented in Figure 1 and their physical properties and chemical compositions are shown in Table 1 and Table 2.

  1. Cement:

Using the Portland Type I cement produced by the Taiwan Cement Corporation and the properties of this cement conformed to ASTM C150, the specific gravity was 3.15, and the fineness was 3,450 cm2/g.

  1. Waste polyethylene (waste PE):

The waste PE its original shape was a large spherical particle. After being broken down by the grinder, its shape was plastic cotton fiber, as shown in Figure 1(b), with a specific gravity of 0.923 and a moisture content was 8.2%. The chemical composition of PE is (C10H8O) n. The Fourier-transform infrared spectroscopy (FTIR) spectrum of waste PE showed in Figure 2.

  1. Ground-granulated blast furnace slag (GGBFS):

It was obtained from the CHC Resources Corporation, and the properties conformed to CNS12549, with a specific gravity of 2.9 and a fineness of 4,000 cm2/g.

  1. Fine aggregate:

The fine aggregate was river sand from the Ligang River, and the specific gravity was tested according to ASTM C127. The specific gravity was 2.65, and the water absorption was 1.48%.

2.2. Test specifications and material mix proportions

The fresh properties of cement mortar included slump and flow. For maintenance, the cement mortar samples were made and placed in saturated lime water. In addition, the engineering properties and durability characteristics were studied at ages 3, 7, 28, 56, and 91 days. Due to the difference in density and volume of waste aggregate and natural aggregate, using a weight replacement ratio is unsuitable for concrete mixes in which waste aggregate and natural aggregate are mixed. Weight substitution does not fit the concept of packing density. Therefore, we replaced all waste materials added in this study by volume replacement rate instead of weight replacement rate and calculated their equivalent weight. List the equivalent weights of all materials in Table 1. The unit weight of the material mix ratio is shown in Table 2. The test methods and specifications of this study are shown in Table 3.

Point 9:

About 3.1 Slump section…

Response 9:

Thanks for your advice. It was corrected as indicated.

Since the waste PE has a high water-containing ability and can accommodate the free water of mortar in PE pores, the slump would decrease when the added volume of waste PE increased.

Point 10:

About the grammatical construction of 3.1 Slump

Response 10:

Thanks for your advice. It was corrected as indicated.

It should be …

The slump was 2.2 cm, 2.5 cm, and 2.8 cm when the GGBFS replaced cement proportion was 0%, 10%, and 20%, respectively. As seen, with the increase in the replaced volume of GGBFS, the slump of cement mortar increased. Due to the slower hydration effect of GGBS in the early stage, the initial setting of mortar time is longer. The free water of the cement mortar increased, resulting in the slump of the mortar increased. When the replaced volume of GGBFS was more, the mortar's workability improved. In conclusion, the increase of waste PE addition resulted in the decrease of mortar slump; the increase of mortar workability with the increase of GGBFS replacement amount.

Point 11:

About 3.5 Flexural strength section…

Response 11:

Thanks for your advice. It was corrected as indicated.

After modification, it looks like this:

The experimental results show that when the waste PE content is less than 2%, the percentage of strength reduction will be within 5%. This way, the waste can be effectively recycled and energy consumption and environmental pollution can be reduced.

Point 12:

It is a must to explain earlier in the text what the relation between ultrasonic velocity and quality of concrete is. Ultrasonic velocity is an indicator, not a property of concrete.

Response 12:

Thanks for your advice. It was corrected as indicated.

After modification, it looks like this:

The ultrasonic velocity measurement method will not cause any damage to the cement products, and the simple operation is a method that the engineering field is willing to adopt. The measured wave velocity value is closely related to the compactness of the cement products. Generally speaking, the pulse wave velocity is proportional to the square root of the elastic modulus of concrete and inversely proportional to the square root of the concrete density. The elastic modulus of concrete is proportional to the square root of the compressive strength according to the ACI 318 recommendation, and the pulse wave velocity is proportional to the fourth root of the compressive strength. That means that when the compressive strength of concrete increases with the increase of materials age, the pulse wave velocity will also increase slightly. But at a later curing period of material, the pulse wave velocity in the concrete does not change sensitively with the strength.

Point 13:

About 3.9 Resistivity section…

Response 13:

Thanks for your advice. It was corrected as indicated.

After modification, it looks like this:

At present, the resistance value and electroosmotic value of concrete are used to evaluate its durability of concrete. The resistance value measures the conductivity of the concrete surface. The electrical conductivity of concrete materials is carried out by the movement of ions caused by electrolytic reactions in the pores. Concrete with more pore water or a high water-cement ratio has a lower resistivity. The fewer the pores of the concrete, the higher the density, the longer the electrical conduction path, and the higher the resistance value. The resistivity of concrete significantly influences the corrosion rate of steel bars. When the resistance value of concrete is less than 10⸳cm, the probability of corrosion is high. Still, the resistance value of concrete is greater than 50⸳cm, which can significantly reduce the corrosion rate of steel bars in concrete.

Point 14:

Regarding point 3 in the conclusion...

Grammatically incorrect.

Response 14:

Thanks for your advice. It was corrected as indicated.

After modification, it looks like this:

  1. The compressive, flexural, and tensile strengths of mortar decrease with the increase of the volume of waste PE added. Adding 2% volume of waste PE to the mortar can meet the needs of general engineering use and can achieve the effect of waste recycling. In the later stage of mortar curing, due to the pozzolanic reaction with the addition of GGBFS, with the increase of curing time and the increase of replacement amount, the later stage strength of mortar increases obviously. When the GGBFS replacement amount is 20%, the strength of the mortar is the best. This amount of replacement increases the overall strength of the specimens.

For other revised parts, see Revised Manuscript.

Reviewer 2 Report

In a cement mortar with a fixed water-to-binder ratio (W/B) of 0.5, waste polyethylene (PE) was added at a sand 12 volume ratio of 0%, 1%, 2%, 3%, and 4% in this paper. Meanwhile, replace cement with 0%, 10%, and 20% ground granular blast furnace slag (GGBFS), and stir to make cement mortar. There are some issues needed to be clarified before it can be accepted for publication in Appl. Sci.:

 1.     Pls delete these unnecessary and unfamiliar abbreviations from the paper. The authors should explain or annotate the abbreviations at their first use site. It is beneficial to readers who are not familiar with this field. E.g. PE and GGBFS in keywords.

2.     Line 14, We studied the new properties and. Pls do not use the first person in scientific papers.

3.     Lines 40-41, Pls use the latest data on carbon emissions. The introduction also needs to be further strengthened and the data updated. E.g. A novel development of green UHPC containing waste concrete powder derived from construction and demolition waste, Temperature effect on the thermal conductivity of expanded polystyrene foamed concrete: Experimental investigation and model correction. So, you can also find that EP is also widely used in concrete, the authors need to review them in the manuscript.

4.     The manuscript should be very carefully checked to avoid any errors. The language should be checked throughout the text and any grammar mistakes should be corrected. E.g. kΩ-cm? or kΩ⸳cm!

5.     2.3 Test methods and regulations, Please add more detailed test steps. Taking setting time as an example, PE is a weak phase or fibrous phase, which acts as a weak body or an obstacle to the probe of the Vicat instrument, which affects the results of setting time.

6.     Lines 110, 212, 271 and 309, Error! Reference source not found!

7.     At the end of the introduction, it is necessary to clearly identify the goal and tasks for achieving it, and in the conclusions, give a numbered list of tasks solved.

8.     The research contributions of the paper should be articulated more clearly. The abstract is not representative of the content and contributions of the paper. The abstract does not seem to properly convey the rigor of research.

Author Response

Response to Reviewer #2 Comments

applsci-1989274

Title: Effect of adding waste polyethylene and GGBFS on the engineering properties of cement mortar.

applied sciences

Thanks again for your advice and attached is the revised checking list according to your notice.

Point 1:

In a cement mortar with a fixed water-to-binder ratio (W/B) of 0.5, waste polyethylene (PE) was added at a sand 12 volume ratio of 0%, 1%, 2%, 3%, and 4% in this paper. Meanwhile, replace cement with 0%, 10%, and 20% ground granular blast furnace slag (GGBFS), and stir to make cement mortar. There are some issues needed to be clarified before it can be accepted for publication in Appl. Sci.

Response 1:

Thank you very much. Thanks for your suggestion. The discussion of some of the results has further explained the observed trends.

Point 2:

Pls delete these unnecessary and unfamiliar abbreviations from the paper. The authors should explain or annotate the abbreviations at their first use site. It is beneficial to readers who are not familiar with this field. E.g. PE and GGBFS in keywords.

Response 2:

Thank you very much. Thanks for your suggestion. Corrected as indicated.

Removed some unnecessary abbreviations and annotated them from the paper when they were first used. Hoped it helps readers who are not familiar with the field.

Point 3:

Line 14, We studied the new properties and. Pls do not use the first person in scientific papers.

Response 3:

Thanks for your suggestion. Corrected as indicated.

Point 4:

Lines 40-41, Pls use the latest data on carbon emissions. The introduction also needs to be further strengthened and the data updated. E.g. A novel development of green UHPC containing waste concrete powder derived from construction and demolition waste, Temperature effect on the thermal conductivity of expanded polystyrene foamed concrete: Experimental investigation and model correction. So, you can also find that EP is also widely used in concrete, the authors need to review them in the manuscript.

Response 4:

Thanks for your suggestion. Corrected as indicated.

The revisions and additions to this part have been corrected and presented in lines 28 to 41 of the revised manuscript.

  1. Introduction

Concrete is the most widely used man-made material on the planet and is used in all construction…

(See the revised manuscript for details.)

Point 5:

The manuscript should be very carefully checked to avoid any errors. The language should be checked throughout the text and any grammar mistakes should be corrected. E.g. kΩ-cm? or kΩ⸳cm!

Response 5:

Thanks for your suggestion. Corrected as indicated.

The language and grammar in the manuscript have been checked and corrected. And change kΩ-cm to kΩ⸳cm!

Point 6:

2.3 Test methods and regulations, Please add more detailed test steps. Taking setting time as an example, PE is a weak phase or fibrous phase, which acts as a weak body or an obstacle to the probe of the Vicat instrument, which affects the results of setting time.

Response 6:

Thanks for your suggestion. Corrected as indicated.

The test methods and regulations in the article are supplemented in Table 3 of the revised manuscript.

Point 7:

Lines 110, 212, 271 and 309, Error! Reference source not found!

Response 7:

Thanks for your suggestion. Corrected as indicated.

Point 8:

At the end of the introduction, it is necessary to clearly identify the goal and tasks for achieving it, and in the conclusions, give a numbered list of tasks solved.

Response 8:

Thanks for your suggestion. Corrected as indicated.

Point 9:

The research contributions of the paper should be articulated more clearly. The abstract is not representative of the content and contributions of the paper. The abstract does not seem to properly convey the rigor of research.

Response 9:

Thanks for your suggestion. Corrected as indicated.

The Introduction and Conclusions sections have been completely revised. Details are shown in the revised manuscript.

For other revised parts, see Revised Manuscript.

Round 2

Reviewer 1 Report

This time your manuscript is better. You put much effort to correct it. Obviously it could be done in more perfect way. Anyway, let it be. However, you must subject the text to professional support of English. There are still some mistakes. I would suggest to ask for help the Journal in this field. They offer such support.

Reviewer 2 Report

It can be accepted.